

# The link between lymphocyte subpopulations in peripheral blood and metabolic variables in patients with severe obesity

Tania Rivera-Carranza[1], Oralia Nájera-Medina[2], Rafael Bojalil-Parra[2], Carmen Paulina Rodríguez-López[2], Eduardo Zúñiga-León[3], Angélica León-Téllez Girón[4] and Alejandro Azaola-Espinosa[3]

[1] Universidad Autónoma Metropolitana, Coyoacán, México DF, México
[2] Department of Health Care, Metropolitan Autonomous University, Coyoacán, México DF, México
[3] Department of Biological Systems, Metropolitan Autonomous University, Coyoacán, México DF, México
[4] Division of Clinical Nutriology, Hospital General Dr. Manuel GEA González, Tlalpan, México DF, México

Corresponding authors
Oralia Nájera-Medina,
onajera@correo.xoc.uam.mx
Alejandro Azaola-Espinosa,
azaola@correo.xoc.uam.mx

## ABSTRACT

**Background**. Obesity, a public health problem, is a state of metainflammation that influences the development of chronic degenerative diseases, particularly in patients with severe obesity.

**Objective**. The objective of this study was to evidence immunometabolic differences in patients with different degrees of obesity, including severe obesity, by determining correlations between lymphocyte subpopulations and metabolic, body composition, and clinical variables.

**Methods**. Peripheral blood immune cells (CD4+, CD8+ memory and effector T lymphocytes) were analyzed, and measures of body composition, blood pressure, and biochemical composition (glucose, glycated hemoglobin (HbA1c), insulin, C-reactive protein (CRP), and the lipid profile) were carried out in patients with different degrees of obesity.

**Results**. The patients were classified according to total body fat (TBF) percentage as normal body fat, class 1 and 2 obesity, class 3 obesity, and class 4 obesity. The greater the TBF percentage, the more pronounced the differences in body composition (such as a decrease in the fat-free mass (FFM) that is defined as sarcopenic obesity) and the immunometabolic profile. There was an increase of CD3+ T lymphocytes (mainly CD4+, CD4+CD62-, and CD8+CD45RO+ T lymphocytes) and an increase in the TBF percentage (severity of obesity).

**Conclusions**. The correlations between lymphocyte subpopulations and metabolic, body composition, and clinical variables demonstrated the existence of a chronic, low-intensity inflammatory process in obesity. Therefore, measuring the immunometabolic profile by means of lymphocyte subpopulations in patients with severe obesity could be useful to determine the severity of the disease and the increased risk of presenting obesity-associated chronic degenerative diseases.

## INTRODUCTION

Obesity is a disease characterized by an excessive increase in total body fat (TBF), the result of a disequilibrium between energy ingested and energy spent; this disease is distinguished by being chronic and multifactorial (*WHO, 2020*). There are different classes of obesity, and the most objective manner to diagnose and classify obesity is by determining the TBF percentage (*Rosales, 2012*; *Suárez & Sánchez, 2018*). It is determined through bioelectrical bioimpedance (BIA) and can be related to the body mass index (BMI) (*WHO, 1995*; *Okorodudu et al., 2010*; *Fried et al., 2014*), although TBF provides a more exact classification of obesity.

Obesity is an important public-health problem that is increasing worldwide, augmenting morbimortality and diminishing life expectancy as well as quality of life (*Taylor, 2011*). In addition, health costs regarding the treatment of obesity and its comorbidities were estimated at 200 million USD in 2019 alone, without counting the economic losses to the labor market, due to absenteeism, unemployment, and early retirement (*OECD, 2019*).

Adipose tissue (AT) is an endocrine organ that contains various types of cells, such as preadipocytes, adipocytes, fibroblasts, vascular endothelial cells, and immune cells (*Lee et al., 2016*; *Liu & Nikolajczyk, 2019*). Adipocyte hypertrophy and hyperplasia, the release of fatty acids, and the activation of innate and adaptive immune cells in the adipose tissue of individuals with obesity generate diverse inflammatory stimuli. These include the secretion of proinflammatory cytokines and the activation of signaling pathways such as c-Jun N-terminal kinase (JNK), I$\kappa$B kinase beta (IKK$\beta$), and nuclear factor kappa-light-chain-enhancer of the activated immunological cells (NF-$\kappa$B). This activation increases the expression of target genes to produce more proinflammatory cytokines, such as interleukin 6 (IL-6), tumor necrosis factor alpha (TNF-$\alpha$), interferon-gamma (IFN-$\gamma$), monocyte chemoattractant protein-1 (MCP-1), and IL-1$\beta$. The consequent systemic inflammation acts on other organs such as skeletal muscle, the liver, and the vascular endothelium and generates insulin resistance (IR) (*Zatterale et al., 2020*; *Huo et al., 2023*). Therefore, obesity is considered a low-intensity, chronic inflammatory state, which is also known as metainflammation-related obesity (MIOR) (*De Heredia, Gómez-Martínez & Marcos, 2012*). It leads to the development of chronic degenerative diseases such as metabolic syndrome (MetS) (*WHO, 2020*), type 2 diabetes mellitus, systemic arterial hypertension, cardiovascular disease, polycystic ovary syndrome, articular disease, metabolic hepatic steatosis, obstructive sleep apnea, and gastroesophageal reflux disease, among others. Similarly, due to this immunometabolic alteration, obesity also increases the rate of infection and some types of cancer (*Avgerinos et al., 2019*; *Fariñas Guerrero & López Gigosos, 2021*).

Researchers have shown that in people with obesity, there are also changes in the proportion and functionality of lymphocytes at the local level (in adipose tissue), as well as in peripheral blood, and these variations are accentuated by an increase in BMI and AT (*De Heredia, Gómez-Martínez & Marcos, 2012*). Indeed, MIOR in adipose tissue appears to be the principal factor that influences the leukocyte count in peripheral blood. The increase in total lymphocytes amplifies the inflammatory response, which plays a key role in the onset
of obesity-related comorbidities (*Ryder et al., 2014*; *Rodríguez et al., 2018*). We aimed to evidence immunometabolic differences in individuals with different classes of obesity by determining correlations between peripheral lymphocyte subpopulations and metabolic variables.

## MATERIALS & METHODS

We conducted an observational, cross-sectional, and comparative study in 124 adults of both sexes aged >18 years, from whom we determined anthropometric, body composition, arterial pressure, metabolic biochemical, and lymphocyte subpopulations in peripheral blood. All participants signed an informed consent letter, and the study was reviewed and approved by the Ethics and Research Committee of the Hospital Gea González and the Universidad Autónoma Metropolitana-Xochimilco (approval reference numbers: 46-119-2019 and Agreement 7/22.5.2).

The inclusion criteria were: patients with obesity in a protocol for bariatric surgery (class 3 and 4 obesity) of the Hospital Gea González Obesity Clinic, and students and/or personnel of the Universidad Autónoma Metropolitana-Xochimilco, Mexico City (controls with normal TBF and class 1 and 2 obesity), of both sexes. The exclusion criteria were: patients with infection, pregnancy, autoimmune diseases, renal disease and/or cancer, or who were taking anti-inflammatory or immunosuppressant drugs. The elimination criteria were: individuals who desired to withdraw from the study, those with incomplete data, or those with any of the diseases and/or treatments mentioned under the exclusion criteria.

### Anthropometric and body composition measurements

We measured weight and height with a Seca 704s TM scale with a stadiometer (Seca, México). We measured the waist circumference (WC) with a Executive 6FT W606P stainless-steel metric tape measure (Lufkin, Englewood, CO, USA). We followed the standardized protocol of the International Society for the Advancement of Kinanthropometry (ISAK) when taking measurements. We obtained BMI, TBF, fat-free Mass (FFM), and visceral fat (VF) from an InBody 720 body composition analyzer (BioSpace Co., Ltd.). We asked each patient not to engage in intense physical exercise during the 24 h prior to the study and to arrive at their appointment having fasted for at least 4 h. If not, the fluid distribution in the body change and is underestimate or overestimate FFM. We used the TBF percentage and BMI to establish the class of obesity based on the *WHO (1995)* criteria. The classes are defined as follows: class 1 and 2 obesity, a TBF percentage of 25%–34.9% for men and 30%–39.9% for women; class 3 obesity, a TBF percentage of 35%–39.9% for men and 40%–44.9% for women; and class 4 obesity, a TBF percentage of $\geq 40\%$ for men and $\geq 45\%$ for women (Table 1) (*WHO, 1995*; *Okorodudu et al., 2010*; *Fried et al., 2014*).

### Biochemical and arterial blood tests

We collected peripheral blood samples in 5-mL Vacutainer™ tubes (BD, Franklin Lakes, NJ, USA) from the participants after they had fasted for at least 8 h. We measured the following molecules in peripheral blood: glucose, triglycerides (TG), total cholesterol, high-density

**Table 1  Diagnosis of the nutritional state based on body mass index (BMI) and total body fat percentage.**

| BMI (kg/m$^2$) | Interpretation | Percentage of total body fat |
|---|---|---|
| <18.5 | Low body fat | M: <14% <br> F: <15% |
| 18.5–24.9 | Normal or healthy body fat | M: 14%–17% <br> F: 15%–24.9% |
| 25–29.9 | Normal or healthy body fat borderline high | M: 18%–24.9% <br> F: 25%–29.9% |
| 30–34.9 | Class 1 obesity | M: 25%–34.9% <br> F: 30%–39.9% |
| 35–39.9 | Class 2 obesity | |
| >40–49.9 | Class 3 obesity or morbid obesity | M: 35%–39.9% <br> F: 40%–44.9% |
| >50 | Class 4 obesity or superobesity | M: $\geq$ 40% <br> F: $\geq$ 45% |

**Notes.**
M, male; F, female.

lipoprotein cholesterol (HDL-c), low-density lipoprotein cholesterol (LDL-c), glycated hemoglobin (HbA1c), insulin, and C-reactive protein (CRP). We measured glucose, TG, total cholesterol, HDL-c, and LDL-c with an automatized clinical biochemical analyzer iKem YY/T0654-2008 (KONTROLab, Morelia, México). We determined the insulin concentration by using a simultaneous one-step immunoenzymatic assay, HbA1c by using capillary electrophoresis, and CRP by using a high-sensitivity immunoassay of near-infrared fluorescence probes. We determined arterial pressure twice according to the guidelines of the Official Mexican Regulation for the Prevention, Detection, Diagnosis, Treatment, and Control of Systemic Arterial Hypertension (PROY-NOM-030-SSA2-2017).

## Analysis of lymphocyte populations by flow cytometry

We collected samples of peripheral blood in five mL Vacutainer™ tubes (BD). To identify the different cellular subpopulations, we employed a mixture of commercial monoclonal antibodies conjugated with fluorochromes (BD). We utilized the following combinations of conjugated antibodies: control isotype with forward scatter (FSC) determines cell size and side scatter (SSC) determines complexity cell (to identifies by morphology total lymphocytes, monocytes, and granulocytes); FITC-anti-CD3/PE-anti-(CD16+CD56+)/PerCP-anti-CD19 (identifies T lymphocytes, natural killer [NK] cells, and B lymphocytes); FITC-anti-CD4/PE-anti-CD62L/APC-anti-CD3 (identifies activated T lymphocytes with helper functions); FITC-anti-CD8/PE-anti-CD28/APC-anti-CD3 (identifies activated T lymphocytes with cytotoxic functions); FITC-anti-CD45RA/PE-anti-CD45RO/PerCP-anti-CD4/APC-anti-CD3 (identifies naïve helper T lymphocytes and memory T lymphocytes); and FITC-anti-CD45RA/PE-anti-CD45RO/PerCP-anti-CD8/APC-anti-CD3 (identifies cytotoxic naïve and memory T lymphocytes) (Fig. 1S).

We incubated the cells with the combination of antibodies. Subsequently, we applied the lysis solution, washed the cells in phosphate-buffered saline (PBS), and fixed the cells with 1% paraformaldehyde containing 0.1% sodium azide (NaN$_3$). We analyzed the samples in

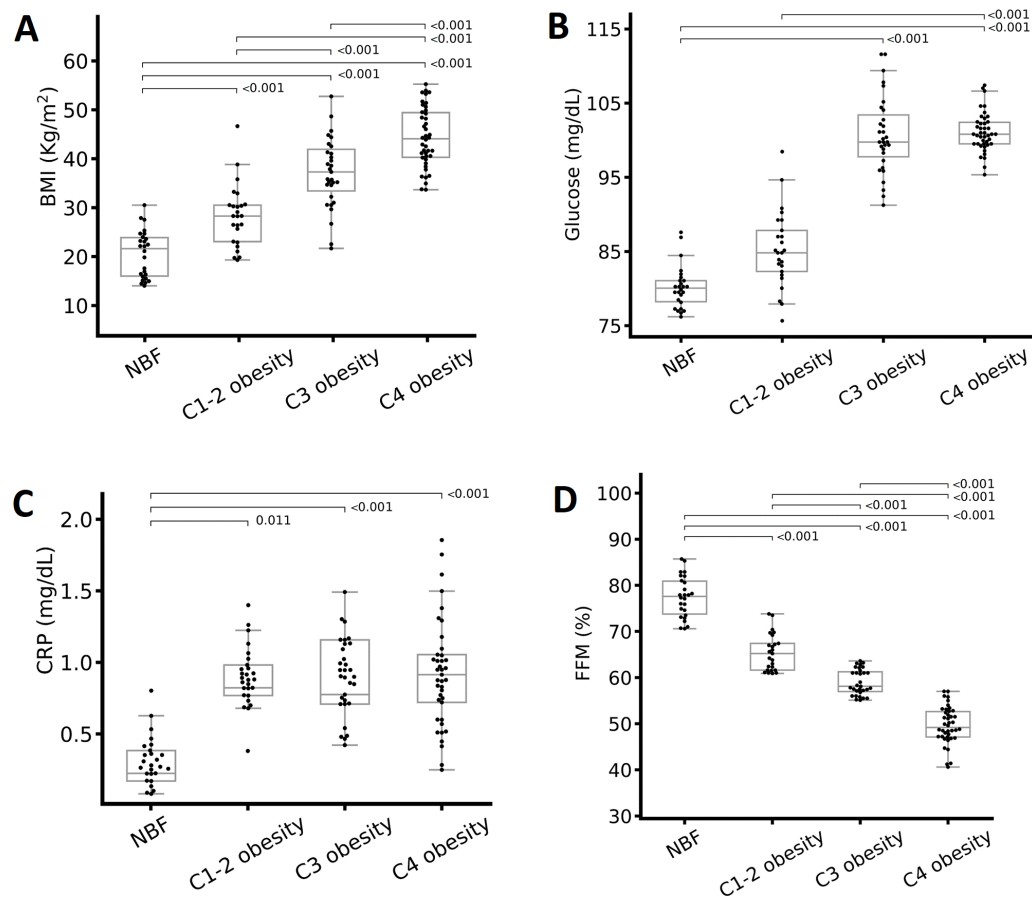

**Figure 1** **Differences in metabolic variables according to the total body fat (TBF) percentage.** (A) BMI, (B) glucose, (C) C-reactive protein (CRP), and (D) fat-free mass (FFM) percentage. The groups were compared with one-way analysis of variance. Abbreviations: C1-2 obesity, class 1 and 2 obesity; C3 obesity, class 3 obesity; C4 obesity, class 4 obesity; NBF, normal body fat. A one-way analysis (ANOVA) of variance was performed to estimate the difference in means. To estimate the difference in means between groups, the Bonferroni post hoc test was used ($p < 0.05$ indicate significant difference).

a flow cytometer (FACScanto TM II; BD) within 24 h of staining. The analysis included a total of 10,000 cells for each event. We used Forward-Scatter and FL-3-Scatter to obtain the percentages of the desired cellular populations and then constructed bi-fluorescent dot plot graphs to delimit regions' lymphocytes subpopulations (*Nájera-Medina et al., 2017*; *Rodríguez et al., 2018*) utilizing FACSDiva version 6.1.3 statistical software.

We calculated the absolute number of each lymphocyte subpopulation as: (the percentage of the required lymphocyte subpopulation × the total number of lymphocytes)/100. The absolute values are expressed in cells/µL.

## Statistical analysis

We analyzed the results after divided the participants into classes of obesity based on TBF percentages. We used the adjusted Kolmogorov–Smirnov method to determine the normality of the data. We transformed the variables that did not pass the normality test to

achieve a normal distribution. The normally distributed data are presented as the mean and standard deviation, and the non-normally distributed variables are presented as the median and interquartile interval. We applied one-way analysis of variance (ANOVA) to each variable to estimate the difference among the groups. We applied two-way ANOVA to each variable to estimate the difference among the groups by sex. We used the Bonferroni post hoc test to determine intergroup mean differences. To determine relationships between lymphocyte subpopulations and metabolic variables, we utilized a Pearson correlation matrix and identified correlations greater than $\pm0.6$. For all tests, we considered $p < 0.05$ to indicate a significant difference. We used the Phyton Environment version 3.6.7 statistical package (CreateSpace 2009, USA) to transform variables and products graphs, and IBM SPSS Statistics version 25.0 (USA) for the remainder of the analyses.

# RESULTS

We evaluated 124 individuals with a mean age of $34.3 \pm 8.8$ years, most of whom were women (64.5%). Overall, 21.0% ($n = 26$) of the participants formed the control group (normal TBF percentage), 20.2% ($n = 25$) had class 1 and 2 obesity, 25.0% ($n = 31$) had class 3 obesity, and 33.8% ($n = 42$) had class 4 obesity. All of the patients with class 3 and 4 obesity had MetS and received treatment for obesity. There were no significant differences in age and sex between the groups (data not shown).

## Metabolic and body composition variables

There were significant differences in most metabolic, body composition, and clinical variables, except for total cholesterol, between the obesity classes. People with class 4 obesity presented higher mean weight, BMI (Fig. 1A), WC, VF, glucose (Fig. 1B), LDL-c (only a significant difference by sex) TG, CRP (Fig. 1C), and systolic blood pressure (SBP). On the other hand, among the obesity classes, people with class 3 obesity presented the highest FFM, HbA1c, insulin, and diastolic blood pressure (DBP), and the lowest HDL-c (Table 2). The normal body fat group presented the highest FFM of all participants (Fig. 1D).

## Immune cell counts

We analyzed the percentages and absolute numbers of cellular populations in peripheral blood for each group (classified based on the TBF percentage). We found significant differences in leukocytes, total lymphocytes, CD19+ lymphocytes (B cells), CD4+ lymphocytes (helper T lymphocytes), CD4+CD62- lymphocytes (effector helper T lymphocytes), CD8+ lymphocytes (cytotoxic T lymphocytes), and CD8+CD45RA+ lymphocytes (naïve cytotoxic T lymphocytes) (Table 3). The percentage of CD19+ lymphocytes and CD8+ lymphocytes decreased as the degree of obesity increased; however, the same did not occur with the absolute numbers of these cells (Table 3). The percentages of CD4+ (Fig. 2A), CD4+CD62- (Fig. 2B), and CD8+CD45RO+ (Fig. 2C) T lymphocytes increased as the degree of obesity increased. The percentage of CD8+CD45RA+ T lymphocytes was lower in individuals with class 3 obesity compared with the rest of the groups (Fig. 2D).

**Table 2  Metabolic, body composition, and clinical characteristics according to the total body fat percentage.**

| Variable (n = 124) | Normal body fat (n = 26) | Class 1 and 2 obesity (n = 25) | Class 3 obesity (n = 31) | Class 4 obesity (n = 42) | p | p@ |
|---|---|---|---|---|---|---|
| TBF (%) | 22.5 ± 4.4 | 34.5 ± 3.9[a] | 40.9 ± 2.7[a,b] | 50.7 ± 4.7[a,b,c] | <0.001 | <0.001 |
| Weight (kg) | 62. ± 12.9 | 77.8 ± 22.2 | 104.4 ± 30.3[a,b] | 117.1 ± 24.6[a,b] | <0.001 | <0.001 |
| BMI (kg/m$^2$) | 22.7 (19.6–24.1) | 28.1 (24.6–29.8)[a] | 37.3 (32.9–42.6)[a,b] | 43.6 (39.5–50.1)[a,b,c] | <0.001 | <0.001 |
| WC (cm) | 78.5 ± 8.8 | 97.4 ± 14.6[a] | 120.2 ± 20.1[a,b] | 129.3 ± 16.0[a,b] | <0.001 | <0.001 |
| FFM (kg) | 48.4 (37.2–57.0) | 47.0 (41.9–54.7) | 55.7 (47.7–78.9)[a] | 53.0 (48.5–60.6)[a] | 0.001 | <0.001 |
| FFM (%) | 77.4 ± 4.4 | 65.4 ± 3.9[a] | 59.0 ± 2.7[a,b] | 49.2 ± 4.7[a,b,c] | <0.001 | <0.001 |
| VF (cm$^2$) | 69.0 (43–95) | 112.6 (98.5–137.7)[a,b] | 173.0 (147.0–277.0)[a,b] | 264 (199–290)[a,b,c] | <0.001 | <0.001 |
| Glucose (mg/dL) | 80.2 (77.6–86.7) | 85.3 (75.5–95.2) | 99 (83.0–113.6)[a] | 103 (95–111)[a,b] | <0.001 | <0.001 |
| HbA1c (%) | 5.4 (5.3–5.5) | 5.4 (5.3–10.5)[a] | 5.8 (5.5–6.7)[b] | 5.7 (5.5–6.0)[b,c] | <0.001 | 0.040 |
| Insulin (μUI/mL) | 6.3 (4.2–7.2) | 14.9 (11.3–31.2)[a] | 22.4 (12.7–27.6)[a] | 21.1 (14.4–30.3)[a] | <0.001 | <0.001 |
| Total cholesterol (mg/dL) | 164.5 ± 27.9 | 171.9 ± 35.8 | 173.3 ± 32.6 | 175.2 ± 31.3 | 0.595 | 0.080 |
| HDL-c (mg/dL) | 49.9 (44.6–60.6) | 43.9 (34.5–53.4) | 37 (33–42)[a] | 41.4 (36.7–48.1)[a] | <0.001 | <0.001 |
| LDL-c (mg/dL) | 86.1 ± 21.6 | 93.4 ± 24.1 | 97.9 ± 28.4 | 103.1 ± 27.8[a] | 0.069 | 0.027 |
| Triglycerides (mg/dL) | 100 (81.8–141) | 128.3 (107.5–184.4)[a] | 138.8 (105.0–186.0)[a] | 150.0 (106.7–195.2)[a,b] | 0.006 | <0.001 |
| CRP (mg/dL) | 0.077 (0.042–0.230) | 0.720 (0.357–0.883) | 0.600 (0.397–1.340)[a] | 0.835 (0.439–1.155)[a] | <0.001 | <0.001 |
| SBP (mmHg) | 106 (100–117) | 110 (107–120) | 120 (110–126)[a] | 125 (110–132)[a,b] | <0.001 | <0.001 |
| DBP (mmHg) | 70 (64–76) | 76 (67–80) | 80 (72–84)[a] | 77 (70–82) | 0.017 | 0.002 |

**Notes.**

The data are presented as mean ± standard deviation or median (interquartile range). Statistical analysis: p, one-way analysis of variance; p@, two-way analysis of variance (adjusted for sex). p < 0.05 is statistically significant.

[a] Significant difference versus individuals with normal body fat (Bonferroni test).
[b] Significant difference versus individuals with grade 1 and 2 obesity (Bonferroni test).
[c] Statistically significant difference versus individuals with grade 3 obesity (Bonferroni test).

## Correlations between cellular populations and immunological, body composition, and metabolic variables

In participants with class 1 and 2 obesity, we found positive correlations between the percentage of CD3+ T lymphocytes and age (Fig. 3A), between the percentage of CD8+CD45RO+ T lymphocytes and WC (Fig. 3B), and between the percentage of CD4+CD45RO+ T lymphocytes and FFM (Fig. 3C), and a negative correlation between CD4+CD45RA+ T lymphocytes and FFM (Fig. 3D).

In the class 3 obesity group, we found positive correlations between leukocytes and VF and CRP (Figs. 4A & 4B) and between granulocytes and CRP (Fig. 4C), and a negative correlation between CD8+CD45RA+CD45RO+ T lymphocytes and VF (Fig. 4D). In the class 4 obesity group, we found positive correlations between the absolute number of CD4+CD45RO+ T lymphocytes and TBF (Fig. 4E), and a negative association between the absolute number of CD8+CD28- lymphocytes and age (Fig. 4F). Considering all participants, we found a positive correlation between the absolute number of CD8+CD45RO T lymphocytes and BMI and WC (Figs. 4G and 4H).
**Table 3  Percentages and absolute numbers of immune cells according to the total body fat percentage.**

| Variable (%, cells/μL) (n = 124) | Normal body fat (n = 26) | Class 1 and 2 obesity (n = 25) | Class 3 obesity (n = 31) | Class 4 obesity (n = 42) | p | p[@] |
|---|---|---|---|---|---|---|
| Leukocytes | 5800 (4900–6675) | 6800 (6100–7642) | 8200 (6400–10800)[a] | 7965 (5825–9825)[a] | **0.013** | **0.001** |
| Monocytes | 7.3 ± 1.8 | 7.7 ± 3.2 | 7.3 ± 2.7 | 7.6 ± 3.0 | 0.916 | 0.233 |
| | 419.5 (332–488) | 418.9 (339–644) | 569.0 (348–889) | 584.5 (383–797)[a] | 0.138 | **0.003** |
| Granulocytes | 65.2 ± 11.4 | 56.9 ± 14.3 | 61.0 ± 11.4 | 63.2 ± 11.4 | 0.090 | 0.337 |
| | 4167.9 (3123–4585) | 4886.0 (4080–5214) | 4349.8 (3804–6511) | 4552.3 (3602–6548) | 0.255 | 0.097 |
| Total lymphocytes | 27.0 ± 12.0 | 34.9 ± 13.5 | 31.6 ± 10.6 | 29.1 ± 11.4 | 0.095 | 0.730 |
| | 1188 (749–1725) | 1464 (844–2010) | 2365 (1613–3630)[a] | 2014 (1498–2704)[a] | **<0.001** | **<0.001** |
| CD16+CD56+ | 22.4 (11.3–30.2) | 12.6 (9.3–18.7) | 19.5 (12.2–25.1) | 18.4 (11.9–25.7) | 0.105 | 0.618 |
| | 308 (188–420) | 182 (145–388) | 467 (397–726)[b] | 332 (223–475) | **0.036** | 0.086 |
| CD3+CD16+CD56+ | 3.9 (1.4–5.4) | 2.9 (2.3–6.7) | 2.5 (1.1–9.8) | 3.5 (1.6–5.3) | 0.984 | 0.836 |
| | 41 (20–101) | 48 (20–93) | 51 (27–174) | 62 (26–103) | 0.809 | 0.357 |
| CD19+ | 14.6 (10.1–20.9) | 9.8 (7.6–15.6) | 9.6 (6.9–11.9) | 9.0 (6.4–12.6)[a] | **0.028** | **0.010** |
| | 154 (114–200) | 172 (125–277) | 237 (115–383)a | 205 (120–350) | 0.310 | **0.043** |
| CD3+ | 59.2 (47.1–66.5) | 77.5 (67.1–79.7) | 73.2 (56.7–78.5) | 68.3 (60.7–79.1) | 0.130 | 0.063 |
| | 716 (390–973) | 952 (543–1384) | 1412 (863–2484)[a] | 1438 (1058–2091)[a] | **<0.001** | **<0.001** |
| CD4+ | 49.4 ± 10.1 | 52.8 ± 11.3 | 58.7 ± 14.2[a] | 59.7 ± 8.8[a] | **0.001** | **<0.001** |
| | 507 (374–904) | 804 (525–1076) | 1122 (870–2213)[a] | 1288 (931–1650)[a] | **<0.001** | **<0.001** |
| CD4+CD62- | 23.6 ± 18.0 | 35.9 ± 12.4 | 37.2 ± 20.4[a] | 44.0 ± 18.7[a] | **<0.001** | **<0.001** |
| | 150 (92–401) | 413 (330–723) | 844 (598–1350)a | 873 (667–1380)[a] | **<0.001** | **<0.001** |
| CD4+CD62+ | 75.8 ± 18.5 | 59.5 ± 13.2[a] | 54.4 ± 21.9[a] | 54.8 ± 18.6[a] | **<0.001** | **<0.001** |
| | 877 (624–1478) | 840 (393–1283) | 1120 (730–2163) | 1164 (694–1883) | 0.258 | **0.009** |
| CD4+CD45RA+ | 23 (13.5–32.5) | 32.4 (21.3–48.9) | 21.9 (14.9–32) | 24.3 (14.3–42.3) | 0.152 | 0.394 |
| | 307 (131–602) | 392 (196–839) | 459 (294–598) | 539 (256–1118)[a] | **0.011** | **0.002** |
| CD4+CD45RO+ | 61.3 (55.8–73.9) | 42.0 (34.7–62.4) | 65.2 (41.4–74.7) | 65.0 (43.1–72.5) | 0.117 | 0.566 |
| | 773 (434–1083) | 773 (414–1215) | 1560 (1219–2497)[a] | 1220 (707–1785)[a] | **<0.001** | **<0.001** |
| CD4+CD45RA+CD45RO+ | 9.6 (8.2–12.8) | 13.2 (8.5–18.7) | 11.9 (7.1–16.9) | 12.8 (8.1–16.5) | 0.523 | 0.324 |
| | 118 (69–141) | 146.6 (102–485) | 209.4 (131–504)[a,b] | 247.4 (153–491)[a,b] | **<0.001** | **<0.001** |
| CD8+ | 56.8 (21.7–65.8) | 36.1 (25.7–41.0) | 31.6 (26.8–43.0) | 29.0 (25.9–34.2)[a] | **0.001** | **<0.001** |
| | 626 (493–859) | 403 (222–866) | 957 (439–1160)[b] | 624 (416–834) | **0.045** | **0.015** |
| CD8+CD28- | 47.6 ± 23.6 | 40.3 ± 16.7 | 45.6 ± 19.1 | 44.0 ± 19.7 | 0.571 | 0.275 |
| | 508 (251–875) | 401 (247–1146) | 955 (505–1660)[a] | 798 (565–1075)[a] | **0.044** | **0.015** |
| CD8+CD28+ | 46.1 ± 21.4 | 57.3 ± 17.4 | 49.6 ± 18.1 | 53.5 ± 19.1 | 0.191 | 0.196 |
| | 338 (249–794) | 580 (515–1348) | 1075 (686–1843)[a] | 1016 (572–1656)[a] | **<0.001** | **<0.001** |
| CD8+CD45RA+ | 44.9 ± 11.9 | 45.8 ± 16.4 | 34.9 ± 14.9 | 35.4 ± 17.3[b] | **0.027** | **0.018** |
| | 495 (373–740) | 455 (218–826) | 702 (487–1107) | 702 (436–1026) | 0.196 | 0.066 |
| CD8+CD45RO+ | 34 (28.3–46.5) | 36.1 (23–43.7) | 45.2 (26.6–64.6) | 46.5 (35.5–70)[a,b] | 0.250 | **0.004** |
| | 392 (246–602) | 638 (438–799) | 931 (736–1587)[a] | 1173 (713–1706)[a,b] | **<0.001** | **<0.001** |

(continued on next page)

**Table 3** (*continued*)

| Variable (%, cells/μL) (n = 124) | Normal body fat (n = 26) | Class 1 and 2 obesity (n = 25) | Class 3 obesity (n = 31) | Class 4 obesity (n = 42) | p | p@ |
|---|---|---|---|---|---|---|
| CD8+CD45RA+CD45RO+ | 16.5 ± 9.4 | 15.1 ± 6.8 | 15.9 ± 9.0 | 15.5 ± 7.2 | 0.840 | 0.794 |
| | 178 (112–311) | 169 (76–516) | 220 (172–381) | 268 (178–580)[a] | **0.039** | **0.011** |

**Notes.**

The data are presented as mean ± standard deviation or median (interquartile range). Statistical analysis: $p$, one-way analysis of variance; $p@$, two-way analysis of variance (adjusted for sex).

[a] $p < 0.05$ is statistically significant

[b] Statistically significant difference versus individuals with normal body fat (Bonferroni test)

[c] Statistically significant difference versus individuals with grade 1 and 2 obesity (Bonferroni test)

Statistically significant difference versus individuals with grade 3 obesity (Bonferroni test)

CD16+CD56+, Natural Killers lymphocytes; CD3+CD16+CD56+, Natural Killers T lymphocytes; CD19+, B lymphocytes; CD3+, T lymphocytes; CD4+, helper T lymphocytes; CD4+CD62+, non-effector helper T lymphocytes; CD4+CD62-, effector helper T lymphocytes; CD4+CD45RA+, naive helper T lymphocytes; CD4+CD45RO+, memory helper Tlymphocytes; CD4+CD45RA+CD45RO+, in transition from naïve to memory helper T lymphocytes; CD8+, cytotoxic T lymphocytes; CD8+CD28+, non-activated cytotoxic T lymphocytes; CD8+CD28-, activated cytotoxic T lymphocytes; CD8+CD45RA+, naive cytotoxic T lymphocytes; CD8+CD45RO+, memory cytotoxic T lymphocytes; CD8+CD45RA+CD45RO+, in transition from naive to memory cytotoxic T lymphocytes.

## DISCUSSION

### Metabolic profile and body composition worsens with the severity of obesity

We conducted this study to identify the immunological, metabolic, and body composition changes in individuals with different classes of obesity, particularly those with severe obesity. In the present work, the anthropometric parameters (weight, WC, and BMI) and body composition (VF) significantly increased according to the TBF percentage increase (Table 2). Only FFM did not demonstrate this pattern: The FFM percentage decreased significantly as the severity of obesity increased (Fig. 1D). These data are consistent with other studies: An increase in obesity is associated with an increase in weight, BMI, WC (*Bauce & Moya, 2019*), and VF (*Valentino et al., 2015*) as well as a decrease in FFM, which indicates sarcopenic obesity (*Barazzoni et al., 2018*; *Poggiogalle et al., 2020*), and an increased risk of presenting comorbidities associated with obesity (*Valentino et al., 2015*).

Regarding the metabolic biochemical and clinical parameters, we observed that the individuals with class 3 obesity had higher mean HbA1c, insulin, and DBP, and lower mean HDL-c (Table 2). The worse metabolic profile in patients with class 3 compared with class 4 obesity could be explained by the fact that individuals with class 4 obesity are asked to lose 5% of their body weight before bariatric surgery (*Hutcheon et al., 2018*). Hence, they adhere better to the dietary recommendations and therefore their metabolic profile improves. In contrast, individuals with class 3 obesity are selected as candidates for bariatric surgery precisely because they have comorbidities associated with obesity and can undergo bariatric surgery sooner, without requiring weight loss. There is an adaptation or metabolic flexibility in individuals with long-term obesity; this could be another reason why individuals with class 4 obesity show a better metabolic profile than individuals with class 3 obesity (*Goodpaster & Sparks, 2017*). Therefore, timely interventions for the diagnosis of comorbidities and for the treatment of obesity (before it becomes severe) are critical to prevent and improve immunometabolic alterations (*Hutcheon et al., 2018*).

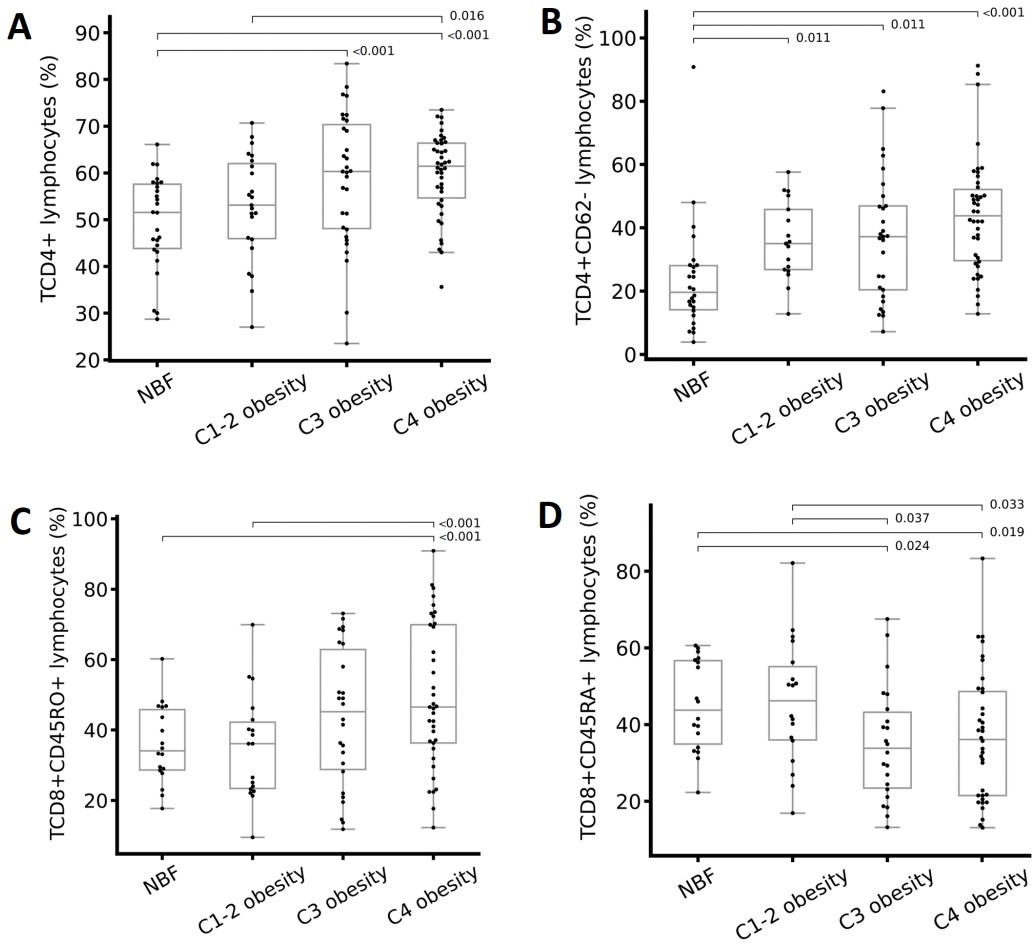

**Figure 2** **Differences in immune cells according to the total body fat (TBF) percentage.** (A) CD4+ T lymphocytes, (B) CD4+CD62- T lymphocytes, (C) CD8+CD45RO+ T lymphocytes, and (D) CD8+CD45RA+ T lymphocytes. Abbreviations: C1-2 obesity, class 1 and 2 obesity; C3 obesity, class 3 obesity; C4 obesity, class 4 obesity; NBF, normal body fat. A one-way analysis (ANOVA) of variance was performed to estimate the difference in means. To estimate the difference in means between groups, the Bonferroni post hoc test was used ($p < 0.05$ indicate significant difference).

## The severity of obesity leads to an increase in the percentages and absolute numbers of T lymphocytes in peripheral blood

T lymphocytes are part of the adaptive immune system. They are produced in the bone marrow and mature in the thymus (lymphoid organ)—hence their name—are identified by the presence of CD3+ on their surface. They are classified into T helper (CD4+), cytotoxic T (CD8+), memory T (CD45RO+), naïve T (CD45RA+), regulatory T (Treg), and other lymphocytes. They are responsible for cellular immunity by destroying infected cells or activating macrophages, CD19+ lymphocytes, or other T lymphocytes by cytokines (messenger proteins responsible for the communication between different cell types) and other costimulatory proteins that are found on their cell membrane (*Dvorkin, Cardinali & Iermoli, 2011*). In individuals with obesity, those previously mentioned factors contribute

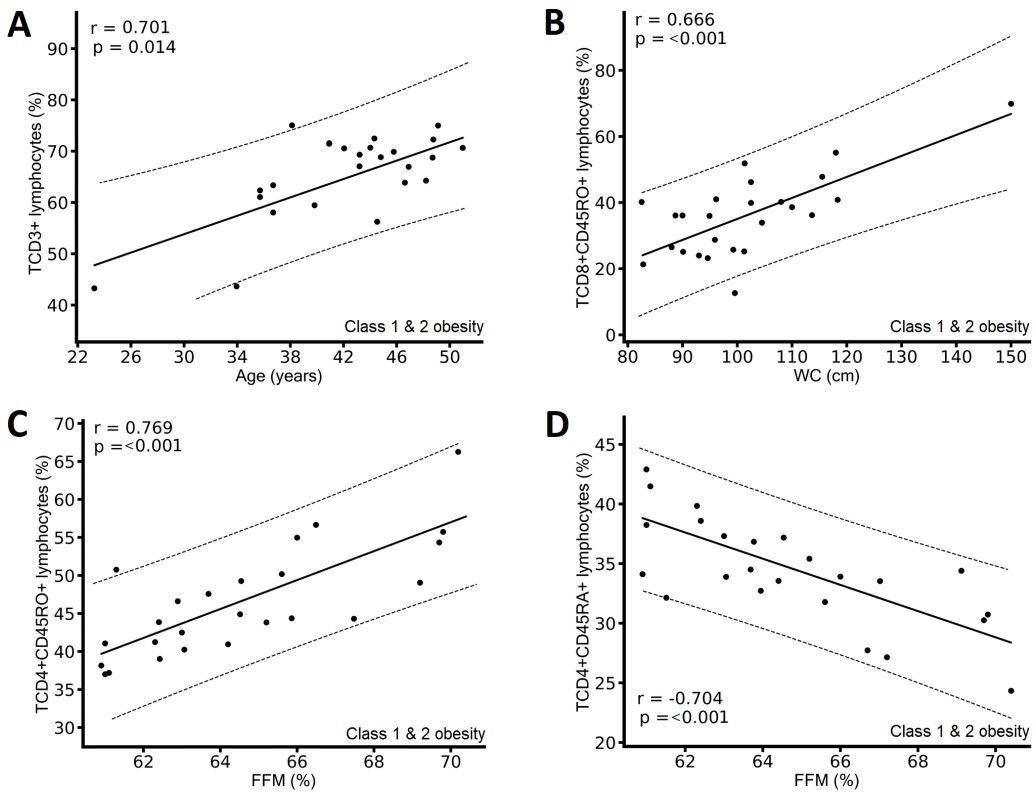

**Figure 3** **Correlation between immunological variables and metabolic and clinical variables of the class 1 and class 2 obesity group.** (A) Positive correlation between CD3+ T lymphocytes and age; (B) positive correlation between CD8+CD45RO+ T lymphocytes and waist circumference (WC); (C) positive correlation between CD4+CD45RO+ T lymphocytes and fat-free mass (FFM); and (D) negative correlation between CD4+CD45RA+ T lymphocytes and FFM. The results show the correlation between two variables calculated with the Pearson correlation matrix, considering correlations (r) greater than ±0.6.

to generate and perpetuate chronic inflammation (*Touch, Clément & André, 2017*; *Schäfer & Zernecke, 2021*).

Effector T lymphocytes, denotes as CD4+CD62-, are a subclass of helper lymphocytes and are involved in the activation of other immune cells, which are particularly important in the adaptive immune response. They are essential to stimulate B lymphocytes to produce antibodies, to activate cytotoxic T lymphocytes, and for increased macrophage activity. In obesity, IR, and MetS, effector T lymphocytes promote the production of proinflammatory T helper cytokines and directly influence obesity-associated inflammation, especially systemic inflammation that causes numerous comorbidities (*Touch, Clément & André, 2017*; *Liu et al., 2022*). We observed that as the degree of obesity increased, CD4+ lymphocytes and CD4+CD62- T lymphocytes also increased (Table 3 and Fig. 2B). This finding is consistent with other studies in which the authors observed an increase in helper T lymphocytes in peripheral blood of individuals with obesity and increased VF (*Rodríguez et al., 2018*) was positive associated with the presence of systemic inflammatory markers such as CRP in plasma and with IR (*McLaughlin et al., 2014*). It is important to point out

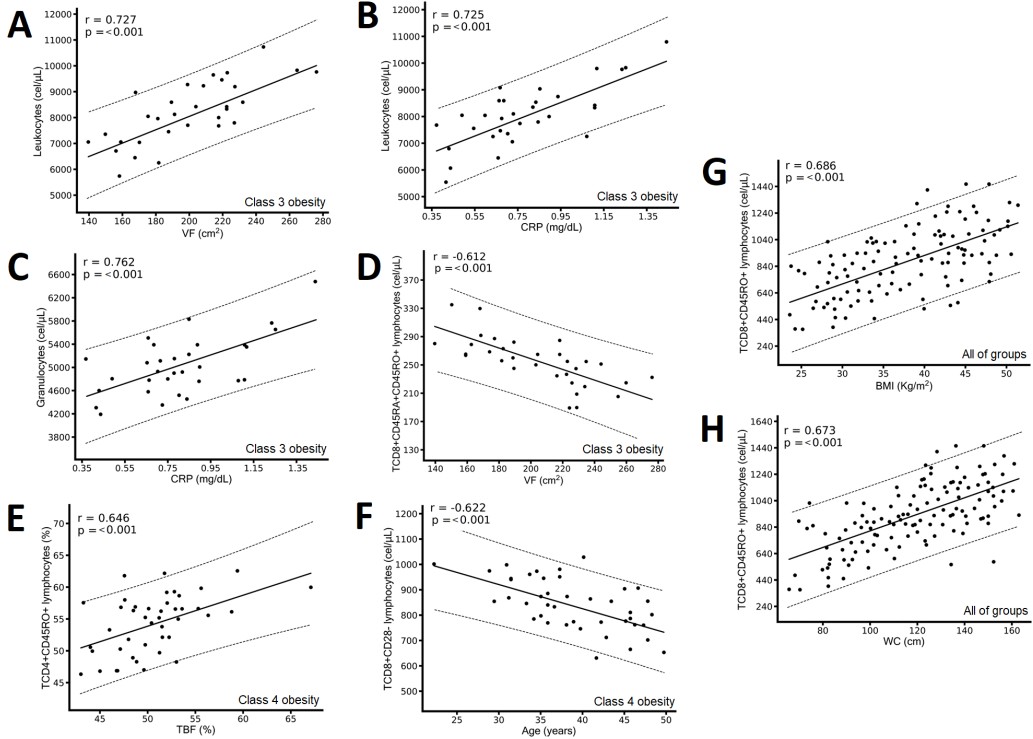

**Figure 4** **Correlation between immunological variables and the metabolic and clinical variables of the class 3 obesity group, the class 4 obesity group, and the entire study group.** The class 3 obesity group has (A) a positive correlation between leukocytes and visceral fat (VF); (B) a positive correlation between leukocytes and C-reactive protein (CRP); (C) a positive association between granulocytes and CRP; and (D) a negative correlation between CD8+CD45RA+CD45RO+ T lymphocytes and VF. The class 4 obesity group has (E) a positive correlation between CD4+CD45RO+ T lymphocytes and total body fat (TBF) and (F) a negative correlation between CD8+CD28- T lymphocytes and age. The entire study group has (G) a positive correlation between CD8+CD45RO+ T lymphocytes and body mass index (BMI) and (H) a positive correlation between CD8+CD45RO+ T lymphocytes and waist circumference (WC). The results show the correlation between two variables calculated with the Pearson correlation matrix, considering correlations (r) greater than ±0.6.

that in these studies, the authors determined obesity based on BMI, unlike our study, in which we classified it based on the TBF percentage. However, the correlation between BMI and TBF in our study is high and significant ($r > 0.8$, $p < 0.001$), so it is possible to compare our results with the aforementioned studies.

Regarding cytotoxic T lymphocytes, we observed that as the degree of obesity increased (TBF), memory T lymphocytes (CD8+CD45RO+) also increased (Table 3 & Fig. 2C) but naïve T lymphocytes (CD8+CD45RA+) decreased (Table 3 & Fig. 2D). We also found the higher the WC and BMI, the higher the peripheral CD8+CD45RO+ T lymphocyte count (Figs. 3B, 4G, & 4H). This finding is in line with other studies: In individuals with obesity and MetS, there are many cytotoxic T lymphocytes and memory T lymphocytes in visceral adipose tissue (VAT) and peripheral blood (*Cancello et al., 2005*; *O'Rourke et al., 2011*; *Anderson et al., 2013*; *Ryder et al., 2014*; *Patel et al., 2023*) there are associated positively with MIOR. Stimulation of T lymphocytes with IL-15 (a proinflammatory cytokine

abundant in individuals with obesity and fatty liver, and who consume a high-fat diet; (*Cepero-Donates et al., 2016*)) increases the expression of the *CPT1A* gene, which, in turn, promotes the oxidation of fatty acids, which are vital for providing energy for cytotoxic T lymphocyte proliferation and survival (*Van der Windt et al., 2013*). Activation of cytotoxic T lymphocytes constitutes one of the first events in the inflammatory response associated with obesity, because it promotes the recruitment and differentiation of macrophages in AT (*Nishimura et al., 2009*). Finally, cytotoxic memory T lymphocytes promote chronic systemic inflammation (*Schäfer & Zernecke, 2021*).

We found that younger patients with class 4 obesity had higher blood levels of CD8+CD28- T lymphocytes (cytotoxic effector T lymphocytes) (Fig. 4F). These lymphocytes are responsible for the effects and functions of cellular immunity; they destroy targeted cells by releasing substances stored in previously performed granules. In addition, they are capable of secreting proinflammatory cytokines such as IFN-$\gamma$. VAT can activate cytotoxic to effector T cells, its infiltration is an early event in the development of the inflammatory response associated with obesity since it promotes the recruitment and differentiation of macrophages, therefore it has essential functions in the initiation and maintenance of the inflammation of the AT and at a systemic level in the development of comorbidities such as IR (*Nishimura et al., 2009*).

In addition, an increase in cytotoxic effector and memory T lymphocytes in peripheral blood is the cause of acute endothelial injury that leads to circulatory diseases, such as atherosclerosis. In individuals with severe obesity, this endothelial damage produced by CD8+ T lymphocytes and is compounded by high concentrations of LDL-c and low concentrations of HDL-c (*Schäfer & Zernecke, 2021*). Finally, it is interesting to mention that the increase, activation, and proliferation of memory T lymphocytes (CD8+CD45RO+, CD4+CD45RO+, and CD8+CD28-) in patients with obesity during chronic inflammation results in the loss of telomerase (memory T lymphocytes with short telomeres), and this in turn leads to decreased naïve T lymphocytes (with long telomeres). These changes condition people with obesity to experience an increased risk of diseases associated with aging and chronic inflammation. This premature aging favors the generation of dysfunctional mitochondria, which leads to the production of reactive oxygen species (ROS) and activation of NF-$\kappa$B that contribute to perpetuate MIOR. For this reason, people with obesity are physiologically and metabolically more likely to age faster than people with normal TBF (*Santos & Sinha, 2021*).

### The greater the severity of obesity, the lower the proportion of FFM, a phenomenon associated with changes in memory and naïve T lymphocytes

In individuals with severe obesity who present sarcopenic obesity, there are also endocrine disorders, premature aging, and decreased physical activity that could generate changes in immune cells (*Kalinkovich & Livshits, 2017*). In the class 1 and 2 obesity group, a decrease in the FFM percentage was associated with a decrease in CD4+CD45RO+ T lymphocytes and an increase in CD4+CD45RA+ T lymphocytes in peripheral blood (Figs. 3C & 3D). Obesity has also been shown to promote increased infiltration of immune cells into

muscle, a phenomenon that might contribute to the chronic low-grade metainflammation associated with obesity (*Patsouris et al., 2014*; *Siervo et al., 2021*). However, more studies are needed to explore this topic.

## Peripheral blood B lymphocytes change with the severity of obesity

We observed that as the TBF percentage increased, the percentage of CD19+ (B lymphocytes) decreased and the absolute number increased in peripheral blood (Table 3). This finding is consistent with the study by *Frasca et al. (2016)* carried out in an American population of 18 young adults between 20 and 40 years old and adults >60 years old, where the percentage of B lymphocytes was lower in individuals with obesity compared with lean individuals. However, in this same study individuals with obesity showed a decrease in absolute number of B lymphocytes compared with lean people (*Frasca et al., 2016*). Similarly, in a population of 169 young Mexican adults, *Rodríguez et al. (2018)* reported a nonsignificant increase in the percentage of total B lymphocytes in peripheral blood of individuals with high VAT. In a population of 40 young American women with obesity (27–55 years old), *Frasca et al. (2021a)* observed that the percentages of naïve B lymphocytes are higher and of memory B lymphocytes are lower in peripheral blood compared with AT. Furthermore, they indicated that B lymphocytes in the blood are metabolically less active regarding the production of proinflammatory substances and in the expression of enzymes of glucose oxidation metabolism than in AT (*Frasca et al., 2021a*). In other studies carried out in adult women between 40 and 55 years of age, *Frasca et al. (2019)* and *Frasca et al. (2021b)* found a higher percentage of senescent B lymphocytes and double-negative B lymphocytes (a subset of B lymphocytes that secrete autoimmune antibodies) in peripheral blood and AT of women with obesity compared with thin women.

It has been demonstrated that as obesity increases, B lymphocytes increase because they are among the first immune cells to infiltrate AT, mainly VAT (*Harmon et al., 2016*; *Srikakulapu & McNamara, 2020*). Once inside AT and plasma of individuals with obesity, B lymphocytes produce proinflammatory mediators that regulate inflammatory T lymphocytes and macrophages and secrete adipocyte-specific autoimmune IgG antibodies (*Frasca et al., 2008*; *Frasca & Blomberg, 2020*). In addition, in obesity and aging, the function of B lymphocytes decreases, a phenomenon associated with deficient responses to infections and vaccines (*Muramatsu et al., 2000*; *Sayegh et al., 2003*; *Frasca et al., 2008*; *Frasca et al., 2016*; *Zhai et al., 2016*).

Our study allows us to formulate two hypotheses as to why, in individuals with obesity, the percentage of B lymphocytes in peripheral blood decreases but the absolute number increases: (1) a large number of B lymphocytes and some of their phenotypic variants may be in AT and (2) the number of T lymphocytes increases considerably as the severity of obesity increases, and thus there is a decrease in B lymphocytes. However, it is important to add that we included people with different degrees of obesity, including many with severe obesity.

## CONCLUSION

We found significant differences in the immunometabolic profile of individuals with different classes of obesity. Compared with the control group, an increase in the severity of obesity based on the TBF percentage is associated with an increase in weight, BMI, WC, and VF; a reduction in FFM (sarcopenic obesity); alterations in metabolic biochemical parameters; and changes in lymphocyte subpopulation counts in peripheral blood. As the TBF percentage (severity of obesity) increases, CD4+, CD4+CD62-, and CD8+CD45RO+ T lymphocytes increase in peripheral blood, demonstrating the existence of an inflammatory process at the peripheral level that is also associated with other variables such as WC, BMI, CRP, leukocytes, and age. Therefore, evaluating the immunometabolic profile in patients with obesity can be clinically useful to assess in a timely manner the risk of presenting inflammatory diseases associated with obesity. However, to be more certain of the behavior of the immune cells of individuals with severe obesity, it is important to continue with these studies to obtain the percentages and absolute numbers of immune cells determine their phenotypic variants, their function, and their correlation with metabolic markers in peripheral blood and AT.

## ACKNOWLEDGEMENTS

The authors thank Martín E. Rojano-Rodríguez MD, General Coordinator of the Obesity Clinic, and Silvia Villanueva-Recillas, Head of the Clinical Laboratory Department, both from the Hospital General Dr. Manuel Gea González, for providing the biological samples.

### Funding

This work was supported by the CONACyT-México via a grant to Tania Rivera-Carranza, MsC (557117). The funders had no role in study design, data collection and analysis, decision to publish, or preparation of the manuscript.

### Grant Disclosures

The following grant information was disclosed by the authors:
CONACyT-México via a grant to Tania Rivera-Carranza, MsC: 557117.

### Competing Interests

The authors declare there are no competing interests.

### Author Contributions

- Tania Rivera-Carranza conceived and designed the experiments, performed the experiments, analyzed the data, prepared figures and/or tables, authored or reviewed drafts of the article, and approved the final draft.
- Oralia Nájera-Medina conceived and designed the experiments, analyzed the data, authored or reviewed drafts of the article, and approved the final draft.

- Rafael Bojalil-Parra conceived and designed the experiments, analyzed the data, authored or reviewed drafts of the article, and approved the final draft.
- Carmen Paulina Rodríguez-López performed the experiments, and approved the final draft.
- Eduardo Zúñiga-León analyzed the data, prepared figures and/or tables, and approved the final draft.
- Angélica León-Téllez Girón conceived and designed the experiments, performed the experiments, prepared figures and/or tables, authored or reviewed drafts of the article, and approved the final draft.
- Alejandro Azaola-Espinosa conceived and designed the experiments, analyzed the data, authored or reviewed drafts of the article, and approved the final draft.

## Human Ethics

The following information was supplied relating to ethical approvals (*i.e.*, approving body and any reference numbers):

The study was reviewed and approved by the Ethics and Research Committee from Hospital Gea González and the Universidad Autónoma Metropolitana. Both institutions approved to carry out the study whitin its facilities (Ethical reference numbers: 46-119-2019 and Agreement 7/22.5). All participants signed an informed consent letter.

## Data Availability

The raw data are available in the Supplemental File.

## Supplemental Information

Supplemental information for this article can be found online at http://dx.doi.org/10.7717/peerj.15465#supplemental-information.

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
