# Peer review of "The link between lymphocyte subpopulations in peripheral blood and metabolic variables in patients with severe obesity"

_PeerJ, doi:10.7717/peerj.15465_

## Round 0.1 · original submission · Minor Revisions

The manuscript is well-written and organized. However, there are a few minor concerns raised by the reviewers which need to be addressed before being considered for publication. The authors are highly recommended to provide the gating strategy for the flow cytometry analysis and maintain uniformity in data representation and statistical method.

Reviewer 1 ·

Basic reporting

NA

Experimental design

NA

Validity of the findings

NA

Additional comments

See attached file

Annotated reviews are not available for download in order to protect the identity of reviewers who chose to remain anonymous.

Reviewer 2 ·

Basic reporting

Rivera-Carranza et al. studied the link between lymphocyte subpopulations in the blood and the metabolic variables in patients with obesity. Overall, the study is well designed and written. The conclusions are adequately derived based on the data provided. Minor suggestions that will improve the manuscript are below.
1. Figure legends for Fig-2, 3 and 4 are not complete. I am not sure if it is a formatting typo?
2. Authors should use uniform labelling for the panels which is conventionally on the upper left corner of each panel.

Experimental design

Well designed and controlled.

Validity of the findings

The data provided support the conclusions.

Reviewer 3 ·

Basic reporting

1: what statistics was used in fig 1 and fig 2? Can the authors send a copy of statics report. Also, to keep in harmony, if the authors are comparing the statistical significance between each of the groups, then they should keep the same trend in all the figures. For example, in fig 1A, all the groups have been compared among themselves and their statistical significance has been shown. I would suggest the authors to follow the same trend in all the figures.
2: A detailed gating strategy should be included for all flow cytometry data.
3: Obesity is characterized by increase in BMI, weight, WC, VF and decrease in FFM percentage. Again TBF is a good indicator to determine obesity as shown by the authors. If so, why did the authors chose to present their data in a different way in fig 3 and fig 4. Why not representing the immunological parameters with respect to TBF? My suggestion will be first representing the observation with respect to TBF and then it will be absolution fine to break it down to further subclasses. This will help in better understanding of the observations to its respective readers. For example: fig 2C shows the CD8+CD45RO+lymphocytes against TBF which gives a fair idea that as obesity progresses, the percentage of CD8+CD45RO+lymphocytes also increase. Now in fig 3B, a correlation has been drawn between CD8+CD45RO+lymphocytes and WC is understandable. For other immunological parameters, I would suggest to first represent with respect to TBF.

Experimental design

NA

Validity of the findings

na

Additional comments

na

·

Basic reporting

This manuscript by Rivera-Carranza and coworkers titled ‘The link between lymphocyte subpopulations in peripheral blood and metabolic variables in patients with severe obesity’ reports comprehensive analysis of lymphocyte subpopulation frequencies in peripheral blood of obese subjects undergoing bariatric surgery. This study is interesting. I appreciate the authors for conducting a comprehensive statistical analysis. Introduction and discussion were appropriately written. Methods were described adequately clear. Results section requires minor improvement in writing. The data was supported by good quality figures.
I ask authors to address the following minor concerns.
1) Please edit the results section to make definitive statements. Please clearly write which cell populations increased/decreased instead of writing ‘there was a link’ or ‘correlated’ or ‘associated’ which makes results description vague. While describing correlation please state if the correlation is positive or negative. Please edit the relevant sentences in abstract and discussion too.
2) In figures, please use ‘greater than’ symbol to denote p values lesser than 0.0001. it is not appropriate to write 0.000
3) In scatter plots showing correlations (figures 3 and 4), please indicate what the dotted lines denote for.
4) Though it is obvious from the figure that the authors performed Pearson correlation, please specify it in methods as the readers of this manuscript may not be biostatisticians.

Experimental design

No comment

Validity of the findings

No comment

Reviewer 5 ·

Basic reporting

none

Experimental design

The authors can shed light on why patients were asked not to engage in intense physical exercise 24 hours prior to the study. The justification for this can enlighten the readers, and reinforce researchers to use such precautions as and when required.

Was the normally distributed data expressed as mean with standard deviation, or as mean with standard error?

Validity of the findings

The authors have conducted a great amount of literature search, and prepared this manuscript. However, they can touch upon some of these recent advances relevant to the topic:

T cells, and their subtypes are also being linked with braind erived neurotrophic factors.
Huo Y, Feng Q, Fan J, et al. Serum brain-derived neurotrophic factor in coronary heart disease: Correlation with the T helper (Th)1/Th2 ratio, Th17/regulatory T (Treg) ratio, and major adverse cardiovascular events. J Clin Lab Anal. 2023;37(1):e24803. doi:10.1002/jcla.24803

Colchicine has been tested for its effects on lymphocyte subpopulations in obese people.
Patel TP, Levine JA, Elizondo DM, et al. Immunomodulatory effects of colchicine on peripheral blood mononuclear cell subpopulations in human obesity: Data from a randomized controlled trial. Obesity (Silver Spring). 2023;31(2):466-478. doi:10.1002/oby.23632

Th17 cells are also being discussed in this regard.
Artemniak-Wojtowicz D, Kucharska AM, Stelmaszczyk-Emmel A, Majcher A, Pyrżak B. Changes of Peripheral Th17 Cells Subset in Overweight and Obese Children After Body Weight Reduction. Front Endocrinol (Lausanne). 2022;13:917402. Published 2022 Jul 6. doi:10.3389/fendo.2022.917402

---

## Round 0.2 · accepted · Accept

The authors have adequately addressed the comments and improved the overall quality of the manuscript, which is ready for publication.

Reviewer 1 ·

Basic reporting

NA

Experimental design

NA

Validity of the findings

NA

Additional comments

NA

Reviewer 3 ·

Basic reporting

All my questions have been addressed.

Experimental design

none

Validity of the findings

none

Additional comments

none

·

Basic reporting

The authors addressed my concerns.

Experimental design

None

Validity of the findings

None